# Intratumoral Microbiota Correlates with AP-2 Expression: A Pan-Cancer Map with Cohort-Specific Prognostic and Molecular Footprints

**DOI:** 10.3390/ijms262311587

**Published:** 2025-11-29

**Authors:** Damian Kołat, Piotr Gromek, Lin-Yong Zhao, Żaneta Kałuzińska-Kołat, Mateusz Kciuk, Renata Kontek, Elżbieta Płuciennik

**Affiliations:** 1Department of Functional Genomics, Medical University of Lodz, 90-752 Lodz, Poland; piotr.gromek@stud.umed.lodz.pl (P.G.); zaneta.kaluzinska@umed.lodz.pl (Ż.K.-K.); mateusz.kciuk@biol.uni.lodz.pl (M.K.); elzbieta.pluciennik@umed.lodz.pl (E.P.); 2Department of Molecular Biotechnology and Genetics, University of Lodz, 90-237 Lodz, Poland; renata.kontek@biol.uni.lodz.pl; 3Department of General Surgery, West China Hospital, Sichuan University, Chengdu 610041, China; 153795352@scu.edu.cn; 4Department of Biomedicine and Experimental Surgery, Medical University of Lodz, 90-136 Lodz, Poland

**Keywords:** microbiota, AP-2, pan-cancer, bioinformatics, prognosis, TFAP2E, TFAP2B, adrenocortical carcinoma, diffuse large B-cell lymphoma, stomach adenocarcinoma

## Abstract

The AP-2 family is a group of key regulators in cancer, yet their relationship with intratumoral microbes remains undefined. The present pan-cancer workflow leveraged TCGA transcriptomic data to correlate expression of AP-2 representatives with bacterial abundance on the genus and species level, followed by cohort-specific survival modeling, clinical profiling, differential expression, weighted co-expression analysis, and chromatin proximity tests with AP-2 enrichment. Significant correlations between microbiota and AP-2 were observed in 18 of 33 analyzed tumor types; *TFAP2E-AS1* was most recurrent among AP-2 members, and Halomonas was most recurrent among genera. Further species-level verification and prognostic importance nominated three promising pairs: *Paraburkholderia fungorum–TFAP2E* in adrenocortical carcinoma (ACC), *Actinomyces oris–TFAP2E* in diffuse large B-cell lymphoma (DLBC), and *Cutibacterium granulosum–TFAP2B* in stomach adenocarcinoma (STAD). An attempt to define a consensus expression signature driven by microbiota and AP-2, yet independent of the specific species or family member, revealed genes regulating various biological processes and pathways. ACC and DLBC shared a consensus expression program, whereas STAD diverged; chromatin analysis showed AP-2 motifs near microbe-responsive genes in ACC and DLBC but not STAD, supporting cohort-specific regulation. Collectively, AP-2 family members emerge as plausible mediators of tumor microbiota–host interplay, warranting further mechanistic and translational research.

## 1. Introduction

The human microbiome, a complex ecosystem of bacteria, viruses, fungi, and protozoa, colonizes diverse anatomical niches (e.g., gut, oral cavity, skin), orchestrating both homeostasis and disease [1,2]. While the majority of research has predominantly focused on the influence of gut microbiota on the efficacy of anti-cancer therapies (particularly immunotherapy) [3,4,5], emerging evidence reveals that microbial communities within non-gastrointestinal tumors actively participate in disease development, progression, and clinical outcomes [6,7]. Advanced sequencing technologies have uncovered distinct microbiomes across cancers, with bacterial composition varying not only between malignancies but also among patients with the same tumor type [8]. Intratumoral bacteria can influence drug metabolism and alter the efficacy of checkpoint inhibitors, highlighting their therapeutic relevance [9,10]. Dysbiosis in tumors correlates with advanced disease stage, metastasis, and poor survival [11,12], suggesting that microbiome-targeted strategies could complement existing cancer therapies.

Microbes are recognized as integral components of the tumor microenvironment, where they modulate inflammation, angiogenesis, and immune evasion through direct microbial-host cell interactions or the secretion of metabolites [13,14,15]. In fact, the regulation of metabolic processes or the immune response is driven by transcriptional mechanisms [16]. The microbiome regulates host gene expression through bidirectional interactions in cellular/extracellular compartments [17]. Microbiome-derived stimuli (e.g., transcripts, metabolites, enzymes, pH changes) are recognized by host cells via extracellular receptors or intracellular entry. These stimuli modulate transcription factor (TF) binding, DNA methylation, chromatin accessibility, transcription, and alternative splicing, driving differential gene/protein expression. Conversely, host-derived proteins mutually shape microbial growth and transcriptional activity, establishing a feedback loop [17]. The microbiota can regulate host gene transcription independently of chromatin accessibility, primarily through differential expression (DE) of TFs and enrichment of their binding sites in nucleosome-depleted cis-regulatory regions [18]. Beyond TF expression, the microbiota also alters TF binding activity and induces epigenetic modifications. Notably, microbial components can directly interact with TFs, as demonstrated for specific host-microbe molecular interfaces [18,19].

For instance, the microbiome inhibits the HNF4α transcription factor, impairing the regulation of inflammatory pathways and promoting pro-inflammatory states [19]. Similarly, the microbiota modulates key transcription factors (e.g., KLF4, AP-1, SP-1), underscoring systemic microbial-TF interactions [20]. Interestingly, one of the TF families important in carcinogenesis, but yet to be explored in relation to the microbiota, is the Activating enhancer-binding protein-2 (AP-2). Microbiome-related research on AP-2 members should be initiated; one of the premises behind this is *H. pylori* infection, which has been shown to disrupt cellular processes by activating the AP-1 family, a member of the same superclass as the AP-2 family [21,22]. Our recent works highlight the broader role of the AP-2 TFs in cancer and gastroenterological disorders, including gastric/colorectal tumors, metabolic diseases, and gut dysmotility [23,24]. The AP-2 family regulates embryogenesis (e.g., limb, eye, and facial development) under homeostatic conditions but exhibits dysregulated functionality in cancer, serving as a prognostic marker [25,26]. AP-2 encompasses five proteins, i.e., AP-2α, AP-2β, AP-2γ, AP-2δ, and AP-2ε, which are encoded, respectively, by *TFAP2A*, *TFAP2B*, *TFAP2C*, *TFAP2D*, and *TFAP2E* genes. In addition, three genes affiliated with the long non-coding RNA (lncRNA) class have been identified, i.e., *TFAP2A-AS1*, *TFAP2A-AS2*, and *TFAP2E-AS1* [27,28]. Structurally, AP-2 proteins belong to the basic Helix-Span-Helix class (bHSH) of Superclass-1, with a transactivation domain at the amino terminus and DNA-binding/dimerization motifs at the carboxyl terminus [27]. The HSH motif facilitates homo- and heterodimerization but requires post-translational modifications (phosphorylation, sumoylation) and interactions with other proteins to modulate subcellular localization, DNA binding, or degradation [25,29]. AP-2 TFs bind GC-rich DNA motifs, with all members except AP-2δ containing a proline-rich activation domain [30]. Previously, we indicated that AP-2 members regulate biological processes that underlie cancer hallmarks [31]. Moreover, these TFs are known to affect the immune system and the effectiveness of related therapies [32], as well as invasiveness [33] or proliferation [23], i.e., aspects frequently discussed in the microbiota literature [11,34]. The importance of AP-2 in the diagnosis of selected tumors, its implication in a network of long non-coding RNAs and microRNAs, as well as the prognostic significance of its targets, certifies that the complexity of the entire family is worth investigating [27,35,36].

As mentioned above, no study has investigated the potential relationship between intratumoral microbiota abundance and AP-2 expression, which could lay the groundwork for future mechanistic research on relevant tumor types. Thus, we present the first analysis of the relationships between microbial abundance and the expression of AP-2 family members, which was preceded by a general pan-cancer assessment of bacterial composition and diversity. By integrating host-microbiome data across a human pan-cancer spectrum, we identified significant correlations between bacterial genera/species and AP-2 family members, as well as evaluated their prognostic and clinical relevance. Furthermore, gene expression patterns associated with these relationships were examined through differential expression and co-expression network analyses. By bridging microbiome and transcription factor research, this work reveals the biological outcomes underlying the microbiota—AP-2 relationship, which may influence cancer development and progression.

## 2. Results

### 2.1. Despite Each Tumor Type Having Its Own Bacterial Composition, Some Commonalities Were Noted

Prior to the investigation focused on the AP-2 family, each cohort was summarized in terms of its bacterial composition (Figure 1). Although heterogeneity between tumors or between patients within a single cohort was expected, it is worth noting that the most prevalent genus in a given cohort also predominates in others, e.g., *Bacillus* is the foremost genus in BRCA, KIRC, LUAD, LUSC, OV, PCPG, PRAD, SKCM, THCA, and UCEC. Other prevailing genera are *Pseudomonas* (CHOL, KICH, LGG, PADD, TGCT, THYM, UVM), *Paenibacillus* (GBM, BLCA, CESC, KIRP, LIHC), *Prevotella* (ESCA, HNSC, STAD), *Bacteroides* (COAD, READ), and *Actinoplanes* (DLBC, MESO). In contrast, the most prevalent genera in ACC, SARC, and UCS are, respectively, *Acinetobacter*, *Peptoclostridium*, and *Saccharomonospora*. In some tumors, mutual exclusivity in bacterial composition can be observed, particularly in KIRP and LIHC, and to a lesser extent in DLBC and SKCM.

Cohorts were also investigated for their microbiota abundance in a pan-cancer spectrum, as well as in terms of diversity indices (Figure 2). The top 50 most prevalent genera included those mentioned earlier (*Bacillus*, *Pseudomonas*, *Paenibacillus*, *Prevotella*, *Bacteroides*, and *Acinetobacter*), as well as bacteria that are less noticeable when analyzing each tumor separately, such as *Escherichia, Klebsiella, Yersinia*, and *Vibrio*. A prevalent characteristic of the tumor microbiome is a subtle decline in the Shannon index compared to control tissues. This finding is often accompanied by a decrease in the Simpson index, indicating domination by a few taxa. Still, some tissues exhibit similar interquartile ranges, independent of the sample type, suggesting that certain tumors have a microbial composition resembling that of normal specimens. As for richness, some tumors, such as BRCA, COAD, OV, and READ, harbor two to three times more taxa than others.

### 2.2. Correlation of Microbiota Abundance and AP-2 Expression Revealed That More than Half of the Cohorts Contain Significant Relationships, with Some of Them Having Prognostic Importance

Focusing on the correlation between bacterial genus abundance and AP-2 family member expression, potential associations were observed in more than half of the cohorts, i.e., 18 of 33 tumor types (Table 1). Between cohorts, the most duplicative AP-2 family member was *TFAP2E-AS1* (DLBC, ESCA, LAML, MESO, PCPG, READ, SKCM, UCS, UVM), while regarding genera, it was *Halomonas* (CHOL, DLBC, GBM, KICH, LIHC, SARC, SKCM). Others found in at least three cohorts were *TFAP2B* (CESC, CHOL, KICH, SARC, STAD, UCS), *TFAP2D* (GBM, LAML, LIHC, MESO, PCPG, SARC), *TFAP2E* (ACC, COAD, DLBC, KICH, READ), *TFAP2A-AS1* (CHOL, DLBC, KICH, PCPG, READ), *Meiothermus* (CESC, DLBC, MESO), and *Corynebacterium* (DLBC, KICH, UCS). The cohort with the highest number of identified relationships was DLBC, while the fewest significant correlations were identified *ex aequo* in ACC, CESC, COAD, ESCA, LIHC, SKCM, STAD, and UVM.

Evaluating the influence of given *TFAP2*-microbiota pairs on patient prognosis revealed that the following bacterial genera or AP-2 family members affect at least one survival endpoint: (1) *Paraburkholderia* and *TFAP2E* in ACC; (2) *Meiothermus* and *TFAP2B* in CESC; (3) *Sphingomonas* and *TFAP2A-AS1* in CHOL; (4) *Brucella* and *TFAP2E* in COAD; (5) *Actinomyces* and *TFAP2E* or *TFAP2E-AS1* in DLBC; (6) *Halomonas* and *TFAP2E-AS1* in DLBC; (7) *Clostridium* and *TFAP2D* in GBM; (8) *Haemophilus* and *TFAP2C* in GBM; (9) *Vibrio* and *TFAP2A-AS1* or *TFAP2A-AS2* in KICH; (10) *Halomonas* and *TFAP2A-AS2* in KICH; (11) *Halomonas* and *TFAP2D* in LIHC; (12) *Mitsuaria* and *TFAP2E-AS1* in PCPG; (13) *Limnohabitans* and *TFAP2E-AS1* in READ; (14) *Polynucleaobacter* and *TFAP2E-AS1* in READ; (15) *Halomonas* and *TFAP2D* in SARC; (16) *Staphylococcus* and *TFAP2B* in SARC; (17) *Cutibacterium* and *TFAP2B* in STAD; (18) *Solimonas* and *TFAP2E-AS1* in UCS; as well as (19) *Staphylococcus* and *TFAP2E-AS1* in UVM. Corresponding genus-level survival analysis is summarized in Appendix A.

### 2.3. Species-Level Correlation Analysis and Assessment of Prognostic Significance Have Enabled the Selection of Promising Relationships in Four Cohorts

The aforementioned relationships were investigated at the species level of a given microbiota, confirming that the majority of these relationships remained significant (Table 2). Occasionally, a single *TFAP2*-microbiota pair on the genus level yielded two or three pairs for investigation on the species level because specific bacteria were sufficiently abundant in a given cohort. For instance, the correlation between *TFAP2E-AS1* and *Staphylococcus* in UVM yielded two relationships at the species level (incorporating *S. aureus* and *S. epidermidis*), whereas for *TFAP2B* and *Cutibacterium* in STAD, it yielded *C. acnes, C. granulosum*, and *C. modestum*. Some pairs were not found at the species level (Meiothermus and TFAP2B in CESC, as well as Sphingomonas and TFAP2A-AS1 in CHOL), making further analysis impossible. A few relationships, although present, did not meet an established threshold for the correlation coefficient and were thus excluded from subsequent study stages. Excluded relationships were: (1) *Brucella anthropi* and *TFAP2E* in COAD; (2) *Halomonas* sp. *JS92-SW72* and *TFAP2E-AS1* in DLBC; (3) *Haemophilus parainfluenzae* and *TFAP2C* in GBM; (4) *Halomonas* sp. *JS92-SW72* and *TFAP2D* in LIHC; (5) *Polynucleobacter* sp. *Adler-ghost* and *TFAP2E-AS1* in READ; as well as (6) *Halomonas* sp. *JS92-SW72* and *TFAP2D* in SARC.

Subsequently, the impact of promising bacterial species on patient prognosis was evaluated (Appendix A). Since all *TFAP2*-microbiota pairs showed a positive correlation (Table 1 and Table 2), it was decided to select those exerting a significant influence on at least one survival endpoint, with both bacterial abundance and AP-2 expression having the same effect on survival. Only the following five relationships met the requirements: (1) *Paraburkholderia fungorum* and *TFAP2E* in ACC; (2) *Actinomyces oris* and *TFAP2E* in DLBC; (3) *Actinomyces oris* and *TFAP2E-AS1* in DLBC; (4) *Halomonas* sp. *JS92-SW72* and *TFAP2A-AS2* in KICH; as well as (5) *Cutibacterium granulosum* and *TFAP2B* in STAD. Since two relationships involved *A. oris* in KICH (paired with either *TFAP2E* or *TFAP2E-AS1*), it was decided to proceed with *TFAP2E*, given its superior statistical significance and correlation coefficient. Corresponding species-level survival analysis is summarized in Appendix A.

### 2.4. Establishment of Representative Patient Groups Reaffirmed Survival Outcomes and Revealed Distinct Clinical Features

Patients from the four selected cohorts underwent intersection analysis to identify representative groups (with higher or lower microbiota abundance and AP-2 expression), which were subsequently evaluated for their prognostic outcomes (Figure 3). Although no significant results were found in KICH, the remaining three cohorts indicated that individuals with simultaneously higher microbiota abundance and AP-2 expression have unfavorable survival relative to those with concurrently lower abundance/expression, based on at least three significant endpoints.

ACC, DLBC, and STAD patients were further evaluated for their clinical picture, revealing that, among the included characteristics, there were more differences in qualitative than quantitative features (Figure 4). In the first cohort, individuals with simultaneously higher *Paraburkholderia fungorum* abundance and *TFAP2E* expression were more frequently characterized by higher tumor stage, C1A molecular subtype, metastasis, COC3 molecular subtype, and steroid phenotype with proliferation pattern. In DLBC, the group with higher *Actinomyces oris* abundance and *TFAP2E* expression consisted solely of females and tended to have less germinal-center B-cell-like (GCB) expression subtype, although the amount of unavailable data or unclassified expression subtype complicates the inference. Regarding STAD, patients with higher *Cutibacterium granulosum* abundance and *TFAP2B* expression showed more homozygous deletions and were less frequently characterized by MSI molecular subtype or more commonly had microsatellite stability; however, this is inconclusive due to the unavailability of data for some individuals. The remaining clinical features with less evident differences between patient groups are depicted in Appendix A.

### 2.5. An Attempt to Identify a Consensus Expression Profile Dependent on AP-2 and Microbiota Uncovered Genes Regulating Various Biological Processes and Pathways

In addition to clinical picture comparison, it was decided to establish a consensus expression profile across cohorts, related to both microbiota and AP-2, but independent of specific species or family members. Simultaneous evaluation of three cohorts using WGCNA revealed one statistically significant gene module associated with microbiota and AP-2 status (Figure 5A). However, the correlation was less satisfactory than that observed in the joint analysis of ACC and DLBC without STAD (Figure 5B), suggesting that the expression profile changes in STAD are generally distinct from those of the other two tumors, and that some essential genes might not constitute a module. Thus, it was decided to employ DEA between patient groups in a given tumor (Appendix A), followed by the intersection analyses between: (1) WGCNA modules from both approaches, or (2) WGCNA modules and DEA-derived differentially expressed genes. Overlapping significant modules identified 116 genes found in the “tan” module from the three-cohort WGCNA and the “midnightblue” module from the two-cohort approach (Figure 5C). In parallel, intersecting WGCNA and DEA data for ACC and DLBC revealed 223 upregulated and 226 downregulated genes that were considered essential for the manifestation of consensus expression profile (Figure 5D). Comparing the lists of 116 and 223 genes, only 18 were found in the “tan” module from the three-cohort WGCNA (Figure 5E), suggesting that expression changes in STAD are devoid of essential genes to the extent that they only partly resemble the consensus profile found in ACC and DLBC.

To assess biological outcomes of such disparity, the above modules and overlapping genes were functionally annotated (Figure 6 and Appendix A). Based on the “WikiPathways cancer” resource, genes residing in the “greenyellow” module are responsible for, e.g., angiogenesis, Wnt signaling, cell adhesion, and interleukin-1 signaling: phenomena that were not found in other gene sets. In contrast, the “midnightblue” module had mutual annotations with the “tan” module and a list of 116 overlapping genes, although with subtle differences. For instance, all three gene sets were linked to ATM signaling, as well as glycolysis and gluconeogenesis. However, the “midnightblue” module is more closely related to DNA mismatch repair and Notch signaling, rather than non-homologous end-joining and Ras signaling, as indicated in the ontology of the other two gene sets. This suggests that the expression profile alterations across three cohorts might entail a few similar biological outcomes, but differences between STAD patient groups do not involve phenomena unique to the “midnightblue” module specific to ACC and DLBC, e.g., metabolic reprogramming or TP53 regulatory network. Extended functional annotation of the same gene sets based on other ontological resources is summarized in Appendix A.

### 2.6. AP-2 Engagement of Microbiota-Responsive Chromatin in ACC and DLBC Contrasts with a Null Pattern in STAD

Ultimately, it was evaluated whether microbe-responsive genes show AP-2 motif and ChIP-overlap enrichment in nearby accessible chromatin, implying AP-2–mediated regulation of the microbiota-host transcriptional program. Across radii, ACC and DLBC displayed consistent enrichment of AP-2 motif sites that intersect the ChIP union in FG sequences relative to GC-matched BG, with the strongest signal at ±750 kb (ACC OR = 1.281, *p* = 2.4 × 10^−5^; DLBC OR = 1.258, *p* = 7.1 × 10^−6^; empirical one-sided *p* ≈ 1 × 10^−4^). Enrichment remained positive at ±1000 kb but with smaller effect sizes (ACC OR = 1.219, *p* = 1.2 × 10^−4^; DLBC OR = 1.106, *p* = 1.6 × 10^−2^). In STAD, the AP-2 family signal was uniformly absent (OR < 1 and *p* > 0.05 at all radii), indicating cohort-specific regulatory divergence (Table 3). These patterns were further verified (Table 4): overlaps between motif-assigned nearest genes and AP-2 regulon targets stabilized around ~12–14% (ACC, AP-2ε regulon) and ~10–11% (DLBC, AP-2ε regulon), whereas STAD (AP-2β regulon) remained low (~2–3%).

## 3. Discussion

The present study establishes the first-ever characterization of the relationship between intratumoral microbiota abundance and AP-2 transcription factor expression across human neoplasms. Through systematic pan-cancer analysis, significant correlations were identified in 18 of 33 tumor types, with subsequent species-level investigation revealing prognostically relevant relationships in ACC, DLBC, and STAD. These findings pave the way for a novel investigative path in cancer biology, suggesting that microbial communities may influence tumor development through previously unexplored transcriptional regulatory mechanisms related to the AP-2 family.

The identification of *TFAP2E-AS1* as the most frequently correlated AP-2 family member represents an encouraging discovery, particularly given its documented role in microRNA regulation [37]. While direct evidence linking *TFAP2E-AS1* to the microbiota remains absent, the established involvement of microRNAs in host-microbe interactions suggests potential indirect regulatory mechanisms that require further investigation [38,39]. The prominence of *TFAP2E* among frequently correlated family members aligns with its documented pleiotropic functions in gastrointestinal diseases, in which the gut microbiota plays an established pathogenic role [24]. This convergence suggests that *TFAP2E* may serve as a transcriptional mediator of microbiota-host interactions in gastrointestinal malignancies. Though mechanistic insights are still lacking, the location of *TFAP2E-AS1* and *TFAP2E* on the same cytogenetic band (1p34.3) suggests a feedback loop between them and a potential dependency of their expression, which may explain the prominence of both genes among frequently correlated family members. The absence of prior literature on the microbiota-*TFAP2D* relationship, despite correlations observed across six cohorts, underscores the novelty of our findings. Given the limited functional characterization of AP-2δ relative to other family members, these observations may guide future research on this understudied transcription factor. Previous studies have noted the involvement of *TFAP2D* in prostate and lung cancer [35,40], suggesting broader relevance across malignancies.

While the predominance of *TFAP2C* expression in ACC has been previously documented [41], our work represents the first evidence suggesting that *TFAP2E* may also play a role in this tumor type. The association of *TFAP2E* and *P. fungorum* levels with aggressive clinical features (including C1A molecular subtype, COC3 classification, and steroid phenotype) aligns with molecular stratification systems, in which these ACC subtypes are associated with worse outcomes [42]. The DLBC findings are particularly compelling, given the extensive evidence for the prognostic significance of microbiota in this malignancy [43,44,45,46,47,48]. While the presence of *Actinomyces* in DLBC has been documented in the literature [49], its correlation with *TFAP2E* represents a novel observation. The potential association with reduced frequency of GCB subtype carries clinical relevance, as microbiota composition influences treatment response and survival in DLBC patients [44,45]. The increased number of females observed in a group with higher *Actinomyces oris* abundance and *TFAP2E* expression may reflect gender-specific immune-microbiome interactions, consistent with emerging evidence of gender-dependent effects of the microbiota on lymphoma progression [46]. For STAD, our findings conform with substantial literature demonstrating the critical role of microbiota in gastric cancer pathogenesis [50,51,52,53]. Although specific data on *Cutibacterium granulosum* are lacking, the putative relationship observed in our study between microsatellite stability and homozygous deletions suggests involvement in genomic instability mechanisms. Previous studies have indicated that microbiota composition influences genomic stability in gastric cancer [52], and AP-2 family members regulate DNA damage response pathways [54], supporting potential mechanistic connections.

The divergent expression profiles between STAD and ACC/DLBC cohorts provide insights into heterogeneous interactions between AP-2 and microbiota across tumor types. The unique STAD profile, involving *TFAP2B* rather than *TFAP2E*, suggests that different AP-2 family members mediate distinct transcriptional programs in response to microbial stimuli. Remarkably, the consensus between ACC and DLBC persists despite fundamental differences in bacterial species (Paraburkholderia vs. Actinomyces) and tumor origins (endocrine vs. hematologic). This convergence, in which *TFAP2E* was consistently involved in both cohorts and maintained a mutual WGCNA pattern across the specific bacterium, suggests that the identity of the responding AP-2 factor may be a more critical determinant of transcriptional outcomes than either tumor type or bacterial species. These observations, while derived from our most significant findings in only three cohorts with highly refined patient groups, warrant future investigation into whether the primary determinant of microbiota-induced transcriptional changes is not the tumor context or bacterial identity *per se*, but rather the AP-2 family member engaged in response to microbial stimuli. The FG-BG enrichment and gene-set intersections suggest that AP-2—dependent regulatory programs are selectively engaged in ACC and DLBC within the genomic neighborhoods of microbiota-linked genes, while such coupling is not evident in STAD. The stable OR pattern across radii in ACC and DLBC implies that relevant AP-2—associated elements are distributed across hundreds of kilobases rather than confined to immediate promoters, consistent with enhancer-rich landscapes and distal enhancer-promoter communication. The lack of enrichment in STAD suggests that AP-2 occupancy is either less central to gastric tumor regulatory wiring or operates in contexts not captured by the current microbe-associated gene sets.

Apart from the above most essential relationships, others identified in our initial screening could be elaborated using the literature. The correlation between *Haemophilus* and *TFAP2C* in GBM gains context from evidence linking *Haemophilus* to IDH1 status and glioma grade [55]. For *Sphingomonas* and *TFAP2A-AS1* in CHOL, existing literature suggests potential roles of the aforementioned genus in cholangiocarcinoma pathogenesis [56]. The identification of *Staphylococcus* as a correlate of SARC and UVM parallels findings in lung cancer, where *Staphylococcus* levels are notably elevated in tumor tissues compared to healthy controls [57,58]. *Staphylococcus* exhibits bidirectional effects on carcinogenesis, with pathogenic strains potentially modulating immune responses through virulence factors [59]. Importantly, AP-2 transcription factors in lung cancer show significant correlations with immune cell infiltration, including CD8+ T cells, CD4+ T cells, and other tumor-infiltrating immune populations [60], suggesting potential mechanistic links between transcriptional regulation and immune modulation. This is consistent with *Staphylococcus* influencing CD8+ T cell infiltration and the immune microenvironment in breast cancer [61], indicating that *Staphylococcus* may exert universal immunomodulatory functions across multiple malignancies, potentially through interactions with transcriptional programs.

Transcriptomic insights, followed by functional annotation, revealed that microbiota—AP-2 relationships influence key signaling pathways and fundamental cellular processes, such as the DNA damage response and metabolic reprogramming. AP-2α directly regulates transcription of critical DNA repair genes such as *TOP2A, NUDT1, POLD1*, and *PARP1* in hepatocellular carcinoma, thereby facilitating repair of oxidized DNA lesions [62]. AP-2γ inhibits *GADD45B* expression, with overexpressed *TFAP2C* suppressing *GADD45B* and *PMAIP1* at both mRNA and protein levels in non-small cell lung cancer cells [63]. The identification of ATM signaling and DNA repair pathways aligns with emerging evidence that intratumoral bacteria alter genomic stability by inducing DNA damage and mutations [48,64,65]. Regarding metabolic alterations, the microbiota influences tumor metabolic reprogramming, with changes in glycolysis and gluconeogenesis consistent with its established role in regulating metabolic pathways, including fatty acid synthesis and glutamate metabolism [51]. Moreover, AP-2 family members regulate metabolic processes through their downstream targets, with *TFAP2A* shown to increase *EZH2* expression by perturbing the activity of the nucleosome remodeling complex, contributing to the metabolic rewiring characteristic of cancer cells [23]. The divergence in pathway enrichment between cohorts (with STAD showing distinct patterns compared with ACC and DLBC) may reflect differences in the AP-2 family members involved and thus distinct transcriptional programs. For instance, phenomena unique to ACC and DLBC (incorporating *TFAP2E*-related relationships) included Notch signaling and the TP53 network, whereas alternative pathways concerned STAD (in which a *TFAP2B*-related relationship was noted). This specificity has implications for therapeutic targeting, as different AP-2 members may require distinct intervention strategies.

The consistent observation that elevated bacterial abundance and AP-2 expression correlate with unfavorable survival across multiple endpoints (OS, DSS, DFI, PFI) implies clinical utility. These markers could enhance existing prognostic systems, particularly in ACC, where molecular classification already guides therapy [66], and in DLBC, where microbiota profiling shows promise for patient stratification [43,44]. The species-specific nature of identified correlations opens possibilities for targeted interventions. While direct manipulation of intratumoral microbiota remains challenging, the documented effects of microbiota on treatment response in various cancers suggest potential therapeutic avenues [45,47]. The prominence of *TFAP2E* across multiple tumor types positions it as a putative target, particularly given its presence in the most promising relationships identified throughout this study, as well as its role in gastrointestinal malignancies, where microbiome-based interventions are advancing rapidly [24,50].

Some study limitations and key takeaways are worth mentioning. Despite being the first investigation to incorporate both microbiota and AP-2 data, the correlative nature of the analysis cannot establish causality, necessitating experimental validation through sophisticated approaches such as co-culture systems, animal models, and relevant sequencing methods (e.g., Dual-Seq). The genus-level correlations inevitably aggregate heterogeneous species and therefore served only as an initial, broad screening step. It also explains why many PCC values are modest yet significant, which is why only relationships that remained consistent at the species level and in downstream analyses were further interpreted. The high number of significant correlations in some tumors during genus-level analysis (especially in DLBC) may be due to a relatively small cohort size, and these findings require additional verification in larger cohorts in the future. An absence of significant survival effects in KICH, despite initially promising correlations, highlights the importance of multilayered analysis in microbiome studies. Technically, one should consider the potential variability between sequencing methodologies and taxonomic classification systems across databases. In addition, strain-level variations within species, which can dramatically influence host interactions, are challenging to assess in silico. Finally, *TFAP2B/E* ChIP-seq data are not available in public repositories at the time of analysis; thus, results rely on *TFAP2A/C* and their union as a proxy for family binding. This limitation affects all cohorts equally and is unlikely to create a divergence between ACC/DLBC and STAD.

All things considered, this work establishes a foundation for investigating microbiota-AP-2 interactions in cancer biology. Priority areas for future research include: (1) mechanistic validation of identified relationships, particularly the *TFAP2E*-microbiota axis; (2) characterization of AP-2 antisense transcripts, which are mentioned prominently but lack precise functional studies; (3) investigation of indirect mechanisms, such as microRNA-mediated regulation; (4) expansion to additional tumors or cancer subtypes, particularly those with established microbiome involvement; and (5) development of microbiome-based biomarker signatures incorporating AP-2 members for clinical implementation. The potential for AP-2 family members to serve as therapeutic targets mediating microbiota effects warrants investigation. Given the established role of microbiota in modulating immunotherapy response, understanding how AP-2 transcription factors integrate microbial signals could lead to the development of novel therapeutic strategies. The influence of the microbiome on transcriptional programs and host gene expression opens new avenues for precision oncology, enabling integrated profiling of microbial communities and host transcriptional states to guide personalized interventions and improve the selection and effectiveness of combination therapies.

## 4. Materials and Methods

### 4.1. Assessment of Bacterial Composition and Diversity in a Pan-Cancer View

Bacterial composition data across all tumor types in The Cancer Genome Atlas (TCGA) [67] were obtained from the Bacteria in Cancer (BIC) database [68], and the relative abundance of the top 15 genera was visualized in composition barplots. BIC also served as a source of genus-level diversity data, presented as boxplots for all available indices: evenness, richness, Shannon index, Simpson index of diversity, and Simpson reciprocal index. In addition, bacterial composition was assessed across a pan-cancer spectrum for the top 50 genera using The Cancer Microbiota (TCMbio) database [3], which was also employed to compare alpha-diversity (Shannon, Simpson, Observed, Chao1) across tumors and non-tumor specimens.

### 4.2. Acquisition and Processing of Gene Expression and Microbiota Abundance Data

Data for all tumor cohorts available in TCGA (Table 5) were obtained from the Genomic Data Commons repository [69] using the GDCquery() function of the TCGAbiolinks v2.34.1 R-package. Transcriptomic data were downloaded as raw counts generated by the Spliced Transcripts Alignment to a Reference (STAR) protocol, with the 38th major build of the human genome defined by the Genome Reference Consortium (hg38) selected as the reference. Corresponding clinical profiles were merged from both GDC (as part of GDCquery) and TCGA Clinical Data Resource (TCGA-CDR) [70], with the former used to derive general clinical features while the latter utilized as a source of prognostic endpoints, i.e., overall survival (OS), disease-specific survival (DSS), disease-free interval (DFI), and progression-free interval (PFI). Regarding microbiota, genus-level and species-level abundance data of intratumoral bacteria were downloaded for all TCGA tumors from TCMbio, which utilizes the Kraken2 taxonomic classification system in its pipeline [3]. Data were filtered to investigate genera/species present in at least 20% of a given cohort and in patients having at least two non-zero counts. Among individuals with available gene expression and bacterial abundance data, counts were subjected to Trimmed Mean of M-values (TMM) normalization using the edgeR v4.4.2 R-package. Patients with values acquired through the TMM method were investigated in consecutive stages of the study.

### 4.3. Correlation Analysis

The Pearson correlation coefficient (PCC) between bacterial abundance and gene expression of AP-2 family members was assessed using the rcorr() function of the Hmisc v5.2-3 R-package. A threshold of *p* < 0.05 was applied via the which() function of the base v4.4.3 R-package to retain only statistically significant relationships. Since *TFAP2*-microbiota pairs were later subjected to various analyses, a PCC threshold of at least |0.3| was considered acceptable at this stage. Initial analysis focused on the correlation between genus-level abundance and AP-2 expression; however, the scope of the species-level analysis (see Section 2.3) was outlined following survival analysis (described below) and the interpretation of results for promising genera (see Section 2.2).

### 4.4. Survival Analysis

Optimal gene expression or bacterial abundance cutpoints to stratify patients into two groups based on OS, DSS, DFI, and PFI were established using the EvaluateCutpoints tool [71], and corresponding Kaplan–Meier curves were generated using the ggsurvplot() function of the survminer v0.5.0 R-package. As mentioned in the previous section, the scope of the survival analysis for bacterial data at both the genus and species levels depended on the results of the correlation analysis, with genus-level information utilized first (see Section 2.2) and species-level data used afterwards (see Section 2.3). During the genus-level approach, it was acceptable to proceed with relationships in which only one component (i.e., microbiota or an AP-2 member) evidently affected survival, with the other component showing at least one result approaching statistical significance. At the species-level stage, it was decided to select relationships that exerted a significant influence on at least one survival endpoint, with both bacterial abundance and AP-2 expression having the same effect on survival.

### 4.5. Intersection Analysis Followed by Assessment of Prognostic Outcomes

To establish representative groups (with higher or lower microbiota abundance and AP-2 expression), an intersection analysis was performed using the UpSetR v1.4.0 R-package. Such groups of patients in selected cohorts (see Section 2.4) were subjected to extended survival analysis. In addition to Kaplan–Meier curves generated as in the above section, hazard ratios were assessed using the coxph() function of forestmodel v0.6.2 R-package, whereas the proportional hazards assumption in a Cox model was evaluated using the Schoenfeld residual test performed via the ggcoxdiagnostics() function of survminer v0.5.0 R-package.

### 4.6. Investigating Clinical Profile and Performing Differential Expression Analysis (DEA) Among Representative Groups of Patients

Representative groups were compared in terms of the clinical profile acquired from GDC, encompassing both qualitative and quantitative features, with the latter detailed in Section 4.11. The distribution of qualitative data was visualized as filled stackbars using the ggplot2 v3.5.1 R-package, while quantitative information was depicted as beanplots generated using the beanplot v1.2 R-package. The same groups of patients were subjected to differential expression analysis (DEA), which was conducted separately for each selected cohort using the limma v3.62.2 R-package, employing the limma-voom method [72,73]. The workflow incorporated normalization through the calcNormFactors() function and the elimination of lowly expressed transcripts (those with ≥5 counts per million in ≥1 library), followed by variance modeling via the voom() transformation. Model fitting was performed in limma using weighted least squares for individual genes through lmFit(), and log2FC values comparing the case group (characterized by higher microbiota abundance and AP-2 expression) against the control group (characterized by lower microbiota abundance and AP-2 expression) were calculated using the makeContrasts() function with default parameters. Empirical Bayes smoothing of standard errors was applied before identifying differentially expressed genes (defined by *p* < 0.05 and log_2_FC > 0.58 or < −0.58) using the topTable() function.

### 4.7. Bootstrapping and Weighted Gene Co-Expression Network Analysis (WGCNA)

Given the data scarcity in one of the representative groups (only four patients in the DLBC cohort representing higher microbiota abundance and AP-2 expression), this group was subjected twice to the resampling bootstrap method [74,75] to acquire a relevant quantity of samples prior to Weighted gene co-expression network analysis (WGCNA) [76]. The latter was performed on all selected cohorts to obtain consensus modules across tumors using the Biological network reconstruction omnibus (BioNERO) [77]. Read counts were subjected to variance stabilizing transformation [78] via the “vstransform” parameter set to “TRUE” and filtered by variance using “variance_filter” set to “TRUE” with *n*  =  10,000 during the data preprocessing. Consensus modules were identified using the consensus_trait_cor() and consensus_modules() functions, with the latter executed with the correlation method set to Pearson and a module merging threshold of 0.2 for a signed hybrid type of network. The transformation into an adjacency matrix was performed using BioNERO-estimated optimal power via the consensus_SFT_fit() function and with the scale-free topology fitting index (R2) > 0.8. Established gene-containing modules were correlated to a binary trait representing microbiota abundance and AP-2 expression, with samples denoted as “AP-2 ↑ Microbiota ↑” or “AP-2 ↓ Microbiota ↓”.

### 4.8. Overlapping DEA and WGCNA Data Alongside Gene Ontology

Given the expression profile disparity between one of the selected cohorts versus the others (see Section 2.5), both WGCNA modules and DEA-derived differentially expressed genes were overlapped using the draw.pairwise.venn() function of the VennDiagram v1.7.3 R-package. The aforementioned modules and overlapping genes were independently subjected to Over-Representation Analysis (ORA) via the WebGestaltR v0.4.6 R-package employing a set of functional annotation resources: Biological Process noRedundant, KEGG, PANTHER, Reactome, and WikiPathways cancer. After setting the reference at “genome” and redundancy removal at “weighted set cover”, up to the top 15 annotations were acquired for each input list of genes.

### 4.9. Proximity Analysis Around Microbiota-Responsive Genes and FG/BG Set Construction

Following identification of ACC, DLBC, and STAD as the most promising cohorts, it was assessed whether microbe-responsive genes show AP-2 motif and ChIP-overlap enrichment in nearby accessible chromatin, suggesting AP-2–mediated regulation of the microbiota-host transcriptional program. Microbiota-responsive gene sets (utilized to center TSS windows) were derived from various sources (more details in Section A.1). Analyses were performed at the cohort level across genomic windows centered on the transcription start sites (TSS) of these genes using four radii (±250, ±500, ±750, ±1000 kb). Accessible chromatin peak sets for ACC and STAD were obtained from the TCGA Pan-Cancer ATAC compendium [79], and the DLBC-oriented analysis utilized ENCODE GM12878 ATAC peaks [80] owing to the lack of DLBC data in the TCGA compendium. Files were converted to BED3 format prior to processing. TSS coordinates were taken from UCSC refFlat on hg38 [81,82]. For each cohort and radius, ATAC peaks were intersected with TSS-centered windows to define foreground (FG) peak windows near upregulated genes, as well as background (BG) peak windows near genes not upregulated. Foreground windows were length/GC matched to background windows to control base composition. GC content was computed with BEDTools nuc [83,84], and adaptive with-replacement matching (tolerances ~2–10% GC and ±20–100 bp) produced a BG set equal in size to FG.

### 4.10. AP-2 Motif Scanning, ChIP-Seq Integration, and Enrichment Testing

To score AP-2 binding motifs, 300-bp windows centered on peak midpoints (pm150) were converted to FASTA using UCSC twoBitToFa [85] against hg38. Position-weight matrices for *TFAP2B* and *TFAP2E* were taken from HOCOMOCO v13 [86]. Motif scanning was performed with FIMO (MEME Suite) [87] using default background and a per-site *p*-value threshold of 1 × 10^−3^; per-sequence maximum scores and motif-hit BEDs were retained for downstream analysis. AP-2 family binding tracks were compiled from the Cistrome Data Browser v3.0 [88]; ChIP-seq peaks were downloaded, converted to BED3, and harmonized to hg38 (where needed) using UCSC liftOver [89]. No peak sets were available for *TFAP2B* or *TFAP2E*; therefore, analyses relied on *TFAP2A* and *TFAP2C* individually and on their union as a surrogate for shared AP-2 occupancy. Motif-ChIP overlap was quantified by intersecting FG and GC-matched BG motif hits with AP-2 ChIP union peaks (bedtools intersect -u [90]). Enrichment of FG relative to BG was tested with a 2 × 2 Fisher’s exact test and summarized as odds ratio (OR) and *p*-value; directional empirical one-sided *p*-values were computed by Monte-Carlo draws from the hypergeometric distribution (N = 10,000). Lists of genes related to AP-2β and AP-2ε were obtained from various sources (more details in Section A.2) and further used for overlap summaries. Included references were UCSC hg38.2bit and refFlat.txt [91,92].

### 4.11. Statistical Analysis

All analyses were performed in RStudio 2024.12.1 build 563 with R 4.4.3 (Posit Software, Boston, MA, USA). Correlations between microbiota abundance and AP-2 expression were quantified using Pearson’s correlation coefficient. Survival endpoints were evaluated using Kaplan–Meier analysis with data-driven cutpoints and Cox proportional hazards models. For quantitative clinical data, normality of distribution was determined by the Shapiro–Wilk test, followed by an unpaired *t*-test or a Wilcoxon test. DEA between representative patient groups was assessed using the limma–voom framework, as well as integrated with consensus WGCNA and ORA. Enrichment near microbiota-responsive genes was tested using matched foreground and background peak sets. Results with a *p*-value less than 0.05 were considered statistically significant. Details on the implementation of specific methods are provided in the above subsections.

## 5. Conclusions

This study provides the first-ever assessment of the relationships between intratumoral microbiota and AP-2 transcription factors, and their widespread identification suggests a novel regulatory axis in human cancer. Initial screening revealed significant correlations in 18 of 33 tumor types, with *TFAP2E-AS1* emerging as the most frequently correlated AP-2 member, and Halomonas as the most prevalent genus. Other frequently correlated components included *TFAP2B*, *TFAP2D*, *TFAP2E*, *TFAP2A-AS1*, *Meiothermus*, and *Corynebacterium*, with the DLBC tumor cohort exhibiting the highest number of identified relationships. Subsequent species-level analysis and prognostic evaluation identified three most promising relationships: *Paraburkholderia fungorum* and *TFAP2E* in ACC, *Actinomyces oris* and *TFAP2E* in DLBC, as well as *Cutibacterium granulosum* and *TFAP2B* in STAD. Across multiple prognostic endpoints, patients with simultaneously elevated microbiota abundance and AP-2 expression had unfavorable survival compared with those with concurrently lower microbiota and AP-2 levels, which was accompanied by distinct clinical feature patterns. Consensus expression profiling, as well as proximity and enrichment analyses across these three cohorts, has revealed important biological insights. While ACC and DLBC exhibited enrichment of AP-2 motif sites around microbiota-responsive genes and shared a consensus transcriptional profile, STAD displayed uniformly absent AP-2 family signals and showed expression changes that only partially resembled the ACC/DLBC consensus. This divergence manifested functionally, with metabolic reprogramming and TP53 regulation network involvement evident in ACC and DLBC but notably absent in STAD, suggesting that different AP-2 family members (*TFAP2E* versus *TFAP2B*) mediate distinct transcriptional programs in response to microbial stimuli. While the mechanistic landscape remains to be elucidated, our findings position AP-2 family members as potential mediators of microbiota-host interactions in cancer, opening new investigative and therapeutic avenues in the rapidly evolving field of cancer microbiome research.

## Figures and Tables

**Figure 1 ijms-26-11587-f001:**
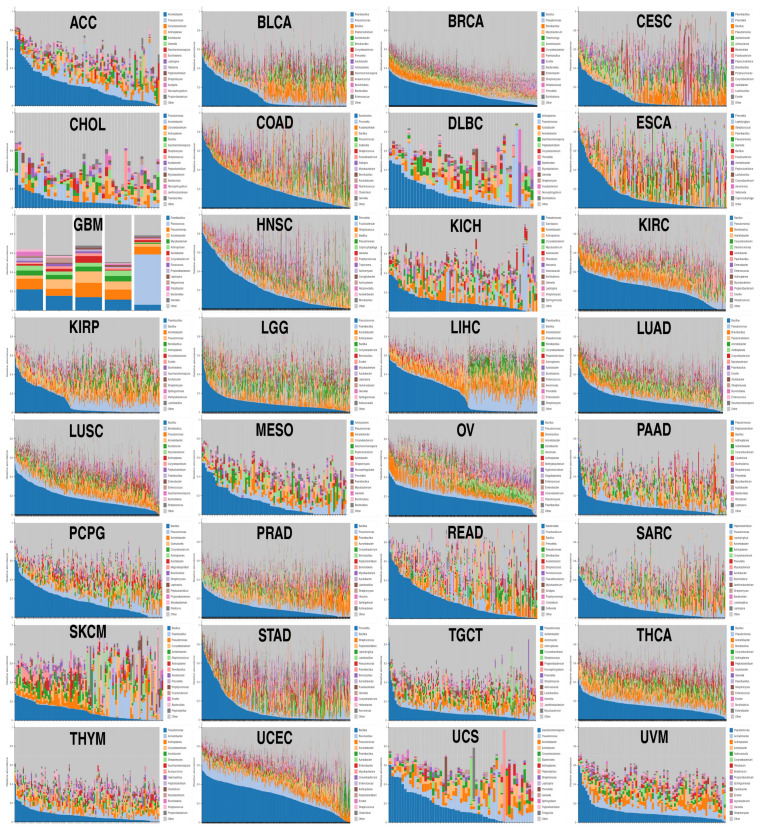
Bacterial composition in tumor cohorts included in the study. The top 15 most abundant genera are specified for each disease.

**Figure 2 ijms-26-11587-f002:**
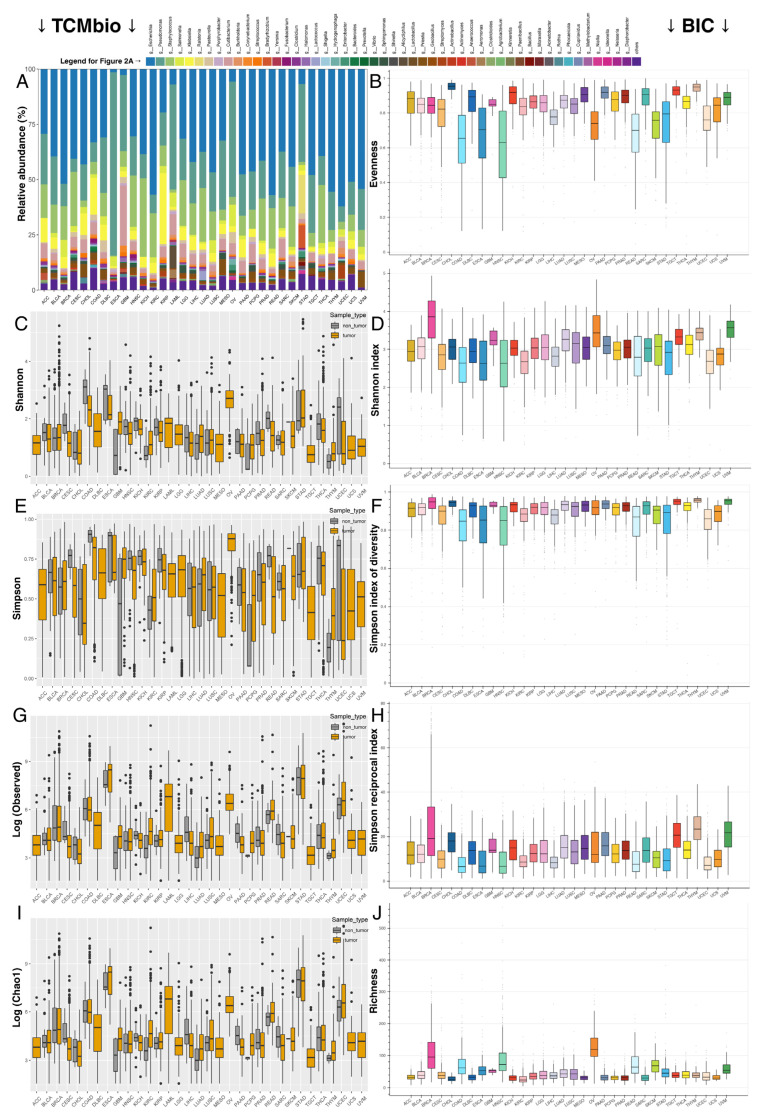
Pan-cancer view of bacterial abundance and diversity indices. (**A**) Top 50 most prevalent genera in a pan-cancer spectrum according to TCMbio data. (**B**) Evenness according to BIC. (**C**,**D**) Shannon indices according to TCMbio and BIC. (**E**,**F**) Simpson indices according to TCMbio and BIC. (**G**) log2-transformed abundances according to TCMbio. (**H**) Simpson’s reciprocal index according to BIC. (**I**) Chao1 index according to TCMbio. (**J**) Richness according to BIC. Data from TCMbio enables comparison between tumors and non-tumor specimens, whereas BIC data focus on differences between tumor types.

**Figure 3 ijms-26-11587-f003:**
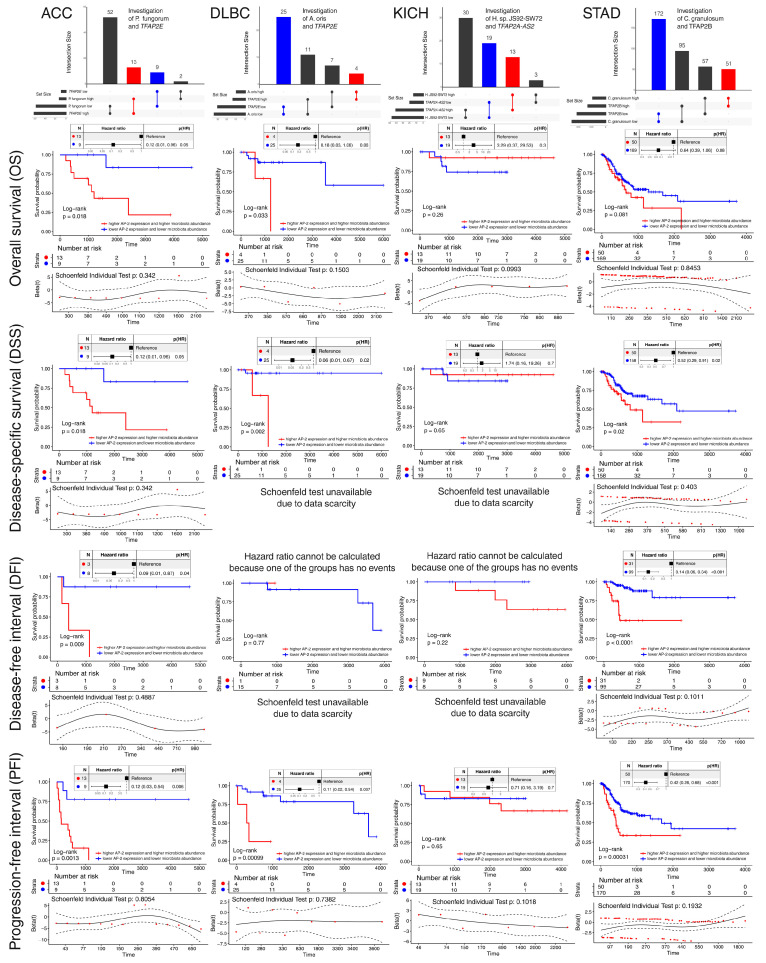
Establishment of representative patient groups in four cohorts with promising *TFAP2*-microbiota relationships, alongside the assessment of their prognostic significance. The upper part depicts the results of intersection analysis, while the remaining part concerns survival analysis, with specific endpoints (OS, DSS, DFI, PFI) in rows and tumor cohorts in columns.

**Figure 4 ijms-26-11587-f004:**
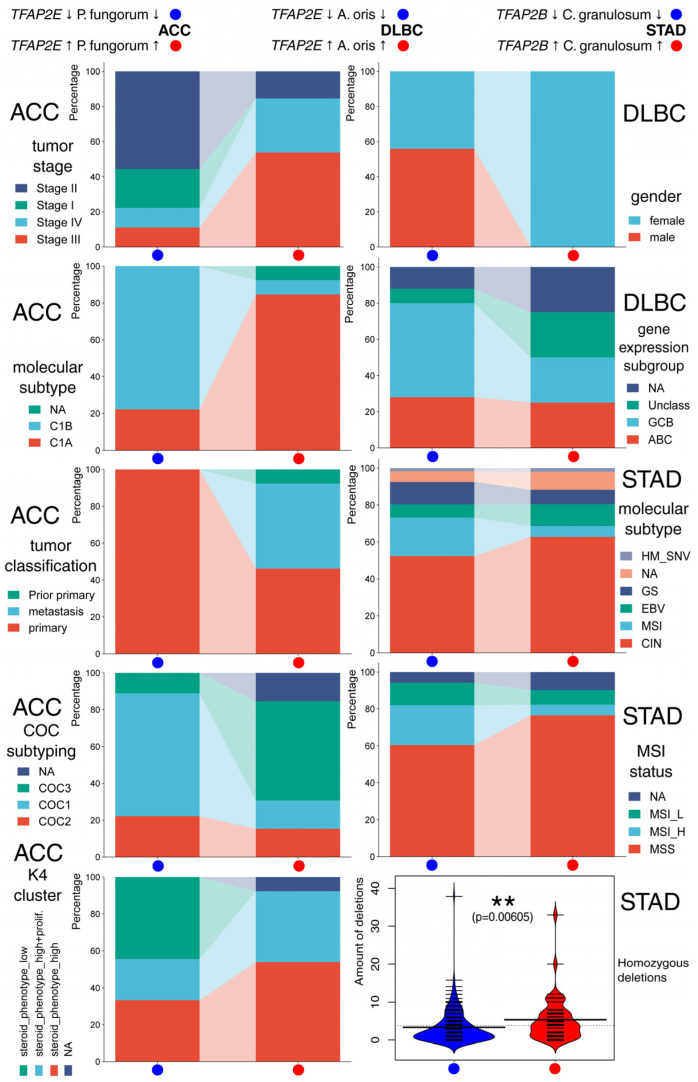
Investigation of clinical differences in representative groups of ACC, DLBC, and STAD patients. Qualitative data were visualized as filled stackbars, while quantitative information was depicted as a beanplot (*p* < 0.01 (**)), with small lines representing individual data points and a larger horizontal line indicating the median per group. Data acquired from the Genomic Data Commons (GDC) portal.

**Figure 5 ijms-26-11587-f005:**
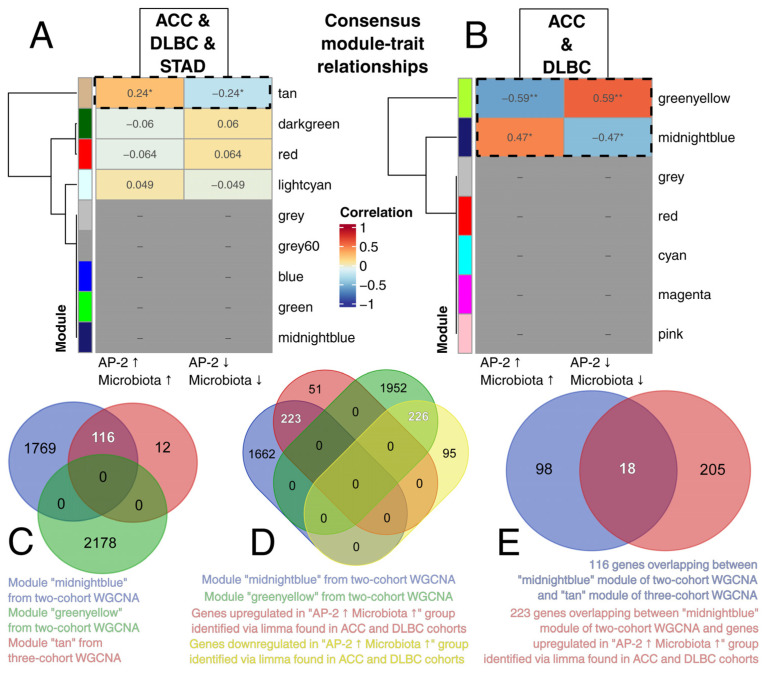
Identification of consensus expression profile dependent on AP-2 and microbiota, with intersection of WGCNA and DEA results. (**A**) Three-cohort WGCNA. (**B**) Two-cohort WGCNA. (**C**) Intersection analysis of significant gene modules from both WGCNA approaches. (**D**) Overlap of WGCNA and DEA data for ACC and DLBC cohorts. (**E**) Identification of a common part between the 116-gene and 223-gene sets found in two previous subpanels. *p* < 0.05 (*), *p* < 0.01 (**).

**Figure 6 ijms-26-11587-f006:**
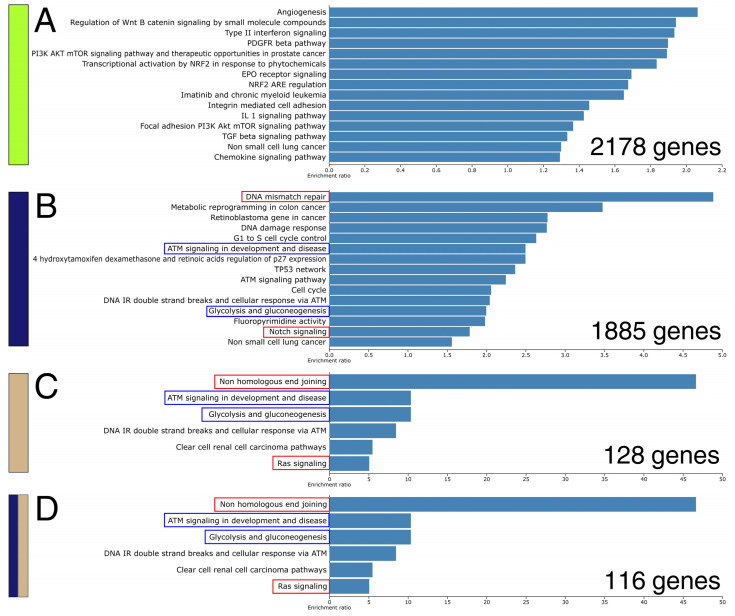
Functional annotation of WGCNA results and genes overlapping between modules according to the “WikiPathways cancer” resource. (**A**) Module “greenyelow” from the two-cohort WGCNA. (**B**) Module “midnightblue” from the two-cohort WGCNA. (**C**) Module “tan” from the three-cohort WGCNA. (**D**) Genes overlapping between “midnightblue” and “tan” modules. The rectangles on the left are colored according to the WGCNA palette. The last subfigure contains a two-color rectangle because it represents the overlap of the “midnightblue” and “tan” modules.

**Table 1 ijms-26-11587-t001:** Correlation between the abundance of bacterial genera and the expression of AP-2 family members in a pan-cancer view.

TCGA Tumor Cohort	Microbiota (Genus-Level)	AP-2 TF	Correlation Coefficient and Statistical Significance
ACC	*Paraburkholderia*	*TFAP2E*	r = 0.55; *p* < 0.0001 (****)
CESC	*Meiothermus*	*TFAP2B*	r = 0.36; *p* < 0.0001 (****)
CHOL	*Halomonas*	*TFAP2B*	r = 0.41; *p* < 0.05 (*)
	*Liberibacter*		r = 0.45; *p* < 0.01 (**)
	*Moraxella*	*TFAP2A-AS1*	r = 0.38; *p* < 0.05 (*)
	*Sphingomonas*		r = 0.35; *p* < 0.05 (*)
COAD	*Brucella*	*TFAP2E*	r = 0.38; *p* < 0.0001 (****)
DLBC	*Actinomyces*	*TFAP2E*	r = 0.65; *p* < 0.0001 (****)
		*TFAP2E-AS1*	r = 0.67; *p* < 0.0001 (****)
	*Alcaligenes*	*TFAP2A-AS2*	r = 0.49; *p* < 0.001 (***)
	*Brucella*	*TFAP2A-AS2*	r = 0.31; *p* < 0.05 (*)
	*Corynebacterium*	*TFAP2E-AS1*	r = 0.33; *p* < 0.05 (*)
	*Cutibacterium*	*TFAP2A-AS2*	r = 0.32; *p* < 0.05 (*)
	*Gordonia*	*TFAP2E*	r = 0.39; *p* < 0.01 (**)
		*TFAP2E-AS1*	r = 0.41; *p* < 0.01 (**)
		*TFAP2A-AS1*	r = 0.31; *p* < 0.05 (*)
		*TFAP2A-AS2*	r = 0.37; *p* < 0.05 (*)
	*Halomonas*	*TFAP2E-AS1*	r = 0.47; *p* < 0.001 (***)
	*Meiothermus*	*TFAP2E*	r = 0.32; *p* < 0.05 (*)
		*TFAP2A-AS1*	r = 0.38; *p* < 0.01 (**)
		*TFAP2A-AS2*	r = 0.51; *p* < 0.001 (***)
	*Paraburkholderia*	*TFAP2E-AS1*	r = 0.36; *p* < 0.05 (*)
		*TFAP2A-AS2*	r = 0.51; *p* < 0.001 (***)
	*Sphingomonas*	*TFAP2A*	r = 0.47; *p* < 0.001 (***)
ESCA	*Shewanella*	*TFAP2E-AS1*	r = 0.38; *p* < 0.0001 (****)
GBM	*Clostridium*	*TFAP2D*	r = 0.79; *p* < 0.0001 (****)
	*Haemophilus*	*TFAP2C*	r = 0.39; *p* < 0.0001 (****)
	*Halomonas*	*TFAP2D*	r = 0.48; *p* < 0.0001 (****)
	*Klebsiella*	*TFAP2D*	r = 0.57; *p* < 0.0001 (****)
KICH	*Corynebacterium*	*TFAP2B*	r = 0.97; *p* < 0.0001 (****)
	*Halomonas*	*TFAP2E*	r = 0.43; *p* < 0.001 (***)
		*TFAP2A-AS2*	r = 0.50; *p* < 0.0001 (****)
	*Vibrio*	*TFAP2E*	r = 0.32; *p* < 0.01 (**)
		*TFAP2A-AS1*	r = 0.72; *p* < 0.0001 (****)
		*TFAP2A-AS2*	r = 0.47; *p* < 0.0001 (****)
LAML	*Aquitalea*	*TFAP2E-AS1*	r = 0.36; *p* < 0.0001 (****)
	*Porphyrobacter*	*TFAP2D*	r = 0.30; *p* < 0.001 (***)
	*Thiopseudomonas*	*TFAP2D*	r = 0.33; *p* < 0.0001 (****)
LIHC	*Halomonas*	*TFAP2D*	r = 0.42; *p* < 0.0001 (****)
MESO	*Meiothermus*	*TFAP2D*	r = 0.35; *p* < 0.01 (**)
		*TFAP2E-AS1*	r = 0.31; *p* < 0.01 (**)
PCPG	*Escherichia*	*TFAP2D*	r = 0.34; *p* < 0.0001 (****)
	*Gordonia*	*TFAP2A-AS1*	r = 0.35; *p* < 0.0001 (****)
	*Mitsuaria*	*TFAP2E-AS1*	r = 0.33; *p* < 0.0001 (****)
READ	*Aquirufa*	*TFAP2E*	r = 0.35; *p* < 0.0001 (****)
	*Moraxella*	*TFAP2A-AS1*	r = 0.32; *p* < 0.0001 (****)
	*Limnohabitans*	*TFAP2E-AS1*	r = 0.34; *p* < 0.0001 (****)
	*Polynucleobacter*	*TFAP2E-AS1*	r = 0.30; *p* < 0.0001 (****)
SARC	*Halomonas*	*TFAP2D*	r = 0.37; *p* < 0.0001 (****)
	*Staphylococcus*	*TFAP2B*	r = 0.38; *p* < 0.0001 (****)
SKCM	*Halomonas*	*TFAP2E-AS1*	r = 0.31; *p* < 0.01 (**)
STAD	*Cutibacterium*	*TFAP2B*	r = 0.51; *p* < 0.0001 (****)
UCS	*Corynebacterium*	*TFAP2B*	r = 0.85; *p* < 0.0001 (****)
	*Priestia*	*TFAP2A*	r = 0.38; *p* < 0.01 (**)
	*Solimonas*	*TFAP2E-AS1*	r = 0.32; *p* < 0.05 (*)
UVM	*Staphylococcus*	*TFAP2E-AS1*	r = 0.50; *p* < 0.0001 (****)

*p* < 0.05 (*), *p* < 0.01 (**), *p* < 0.001 (***), *p* < 0.0001 (****).

**Table 2 ijms-26-11587-t002:** Correlation between the abundance of bacterial species and the expression of AP-2 family members in a pan-cancer view.

TCGA Tumor Cohort	Microbiota (Species-Level)	AP-2 TF	Correlation Coefficient and Statistical Significance
ACC	*Paraburkholderia fungorum*	*TFAP2E*	r = 0.50; *p* < 0.0001 (****)
COAD	*Brucella anthropi*	*TFAP2E*	r = 0.20; *p* < 0.0001 (****)
DLBC	*Actinomyces oris*	*TFAP2E*	r = 0.47; *p* < 0.001 (***)
		*TFAP2E-AS1*	r = 0.46; *p* < 0.01 (**)
	*Halomonas* sp. *JS92-SW72*	*TFAP2E-AS1*	r = 0.29; *p* < 0.05 (*)
GBM	*Clostridium botulinum*	*TFAP2D*	r = 0.32; *p* < 0.0001 (****)
	*Haemophilus parainfluenzae*	*TFAP2C*	r = 0.18; *p* < 0.05 (*)
KICH	*Halomonas* sp. *JS92-SW72*	*TFAP2A-AS2*	r = 0.35; *p* < 0.01 (**)
	*Vibrio anguillarum*	*TFAP2A-AS1*	r = 0.48; *p* < 0.0001 (****)
		*TFAP2A-AS2*	r = 0.39; *p* < 0.01 (**)
LIHC	*Halomonas* sp. *JS92-SW72*	*TFAP2D*	r = 0.18; *p* < 0.001 (***)
PCPG	*Mitsuaria* sp. *7*	*TFAP2E-AS1*	r = 0.38; *p* < 0.0001 (****)
READ	*Limnohabitans* sp. *103DPR2*	*TFAP2E-AS1*	r = 0.37; *p* < 0.0001 (****)
	*Limnohabitans* sp. *63ED37-2*		r = 0.35; *p* < 0.0001 (****)
	*Polynucleobacter* sp. *Adler-ghost*		r = 0.22; *p* < 0.01 (**)
SARC	*Halomonas* sp. *JS92-SW72*	*TFAP2D*	r = 0.25; *p* < 0.0001 (****)
	*Staphylococcus aureus*	*TFAP2B*	r = 0.33; *p* < 0.0001 (****)
STAD	*Cutibacterium acnes*	*TFAP2B*	r = 0.44; *p* < 0.0001 (****)
	*Cutibacterium granulosum*		r = 0.37; *p* < 0.0001 (****)
	*Cutibacterium modestum*		r = 0.63; *p* < 0.0001 (****)
UCS	*Solimonas* sp. *K1W22B-7*	*TFAP2E-AS1*	r = 0.34; *p* < 0.05 (*)
UVM	*Staphylococcus aureus*	*TFAP2E-AS1*	r = 0.38; *p* < 0.001 (***)
	*Staphylococcus epidermidis*		r = 0.72; *p* < 0.0001 (****)

*p* < 0.05 (*), *p* < 0.01 (**), *p* < 0.001 (***), *p* < 0.0001 (****).

**Table 3 ijms-26-11587-t003:** AP-2 family motif-ChIP enrichment across radii.

Cohort	Radius (kb)	Odds Ratio	Fisher *p*-Value	Empirical *p*-Value
ACC	250	1.138	8.915 × 10^−2^	8.139 × 10^−2^
	500	1.156	2.079 × 10^−2^	2.37 × 10^−2^
	750	1.281	2.403 × 10^−5^	1.00 × 10^−4^
	1000	1.219	1.227 × 10^−4^	2.00 × 10^−4^
DLBC	250	1.241	3.496 × 10^−3^	2.90 × 10^−3^
	500	1.167	4.839 × 10^−3^	4.40 × 10^−3^
	750	1.258	7.099 × 10^−6^	1.00 × 10^−4^
	1000	1.106	1.643 × 10^−2^	1.58 × 10^−2^
STAD	250	0.835	8.373 × 10^−1^	2.167 × 10^−1^
	500	0.831	9.068 × 10^−1^	1.195 × 10^−1^
	750	0.827	9.524 × 10^−1^	6.129 × 10^−2^
	1000	0.855	9.378 × 10^−1^	7.909 × 10^−2^

Odds ratios (FG vs. GC-matched BG) and two-sided Fisher *p*-values are shown alongside directional empirical one-sided *p*-values from 10,000 hypergeometric draws.

**Table 4 ijms-26-11587-t004:** Overlap between nearest-gene assignments from motif windows and AP-2 regulon targets.

Cohort	TF	250 kb	500 kb	750 kb	1000 kb
ACC	AP-2ε	84/622 (13.5%)	135/1058 (12.8%)	184/1444 (12.7%)	223/1822 (12.2%)
DLBC	AP-2ε	60/544 (11.0%)	100/939 (10.6%)	137/1255 (10.9%)	163/1538 (10.6%)
STAD	AP-2β	3/134 (2.24%)	8/239 (3.35%)	12/374 (3.21%)	17/469 (3.62%)

**Table 5 ijms-26-11587-t005:** Investigated tumors included in the study, with an indication of each cohort size.

TCGA Cohort Abbreviation	Full Disease Name/Description	Number of Samples Included in This Study, Considering Microbiota Data Availability
ACC	Adrenocortical carcinoma	76
BLCA	Bladder urothelial carcinoma	395
BRCA	Breast invasive carcinoma	1090
CESC	Cervical and endocervical cancers	293
CHOL	Cholangiocarcinoma	33
COAD	Colon adenocarcinoma	456
DLBC	Diffuse large B-cell lymphoma	47
ESCA	Esophageal carcinoma	162
GBM	Glioblastoma multiforme	154
HNSC	Head and neck squamous cell carcinoma	500
KICH	Kidney chromophobe	65
KIRC	Kidney renal clear cell carcinoma	532
KIRP	Kidney renal papillary cell carcinoma	288
LAML	Acute myeloid leukemia	146
LGG	Lower grade glioma	510
LIHC	Liver hepatocellular carcinoma	361
LUAD	Lung adenocarcinoma	496
LUSC	Lung squamous cell carcinoma	495
MESO	Mesothelioma	85
OV	Ovarian serous cystadenocarcinoma	373
PAAD	Pancreatic adenocarcinoma	176
PCPG	Pheochromocytoma and Paraganglioma	179
PRAD	Prostate adenocarcinoma	484
READ	Rectum adenocarcinoma	166
SARC	Sarcoma	253
SKCM	Skin cutaneous melanoma	103
STAD	Stomach adenocarcinoma	375
TGCT	Testicular germ cell tumors	135
THCA	Thyroid carcinoma	502
THYM	Thymoma	116
UCEC	Uterine corpus endometrial carcinoma	545
UCS	Uterine carcinosarcoma	56
UVM	Uveal melanoma	79

## Data Availability

The original contributions presented in this study are included in the article/Appendix A. Further inquiries can be directed to the corresponding author.

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
