# Peer review of "Intratumoral Microbiota Correlates with AP-2 Expression: A Pan-Cancer Map with Cohort-Specific Prognostic and Molecular Footprints"

_ijms, 2025, doi:10.3390/ijms262311587_

Round 1
Reviewer 1 Report
Comments and Suggestions for Authors
Comments;
Minor revision requested
- in Table 1. Correlation between the abundance of bacterial genera and the expression of AP-2 family members in a pan-cancer view ... I see most of the (r) values are very low but significant>> do you have any explanation or interpretation for that?
- Explain the statistical analysis used in this work in the materials and methods.
- Do you have ethical approval for this work?
Author Response
Dear Reviewer 1, thank you so much for all your comments, please see the attachment for our responses. With kind regards, Damian Kołat

Reviewer 2 Report
Comments and Suggestions for Authors
This research paper is well-written and effectively addresses the limitations of the study. However, the authors could enhance the paper by providing the code they utilized. This would allow readers to better understand the methodology and the specific settings employed in the study. A minor issue I encountered is the threshold they have used for the log2FC. I am uncertain about the reason they used an absolute 0.58 filter. An explanation for that choice would be greatly appreciated.
Author Response
Dear Reviewer 2, thank you so much for all your comments, please see the attachment for our responses. With kind regards, Damian Kołat

Reviewer 3 Report
Comments and Suggestions for Authors
In this manuscript, the authors assessed the relationships between intratumoral microbiota and AP-2 expression and demonstrated that AP-2 family members may serve as mediators of tumor microbiota–host interactions. While the manuscript is generally well written, the following points should be addressed before it can proceed to publication.
Table 2:
Six rows contain statistically significant p-values but lack the corresponding asterisks. Please add asterisks to these rows for consistency with the rest of the table.
Figure 4:
For the violin plot positioned at the lower right corner, please provide a more detailed description of this panel in the figure legend. Additionally, clarify how the p-value was calculated.
Figure 6:
The annotations such as “Module ‘greenyellow’ from two-cohort WGCNA” currently placed along the left margin of the figure should be moved to the figure legend. I also recommend adding subpanel labels (A, B, C, D) and adjusting the column colors so that each panel’s color matches the corresponding module color used in Figure 5.
Author Response
Dear Reviewer 3, thank you so much for all your comments, please see the attachment for our responses. With kind regards, Damian Kołat
